# Abscisic Acid Improves Insulin Action on Glycemia in Insulin-Deficient Mouse Models of Type 1 Diabetes

**DOI:** 10.3390/metabo12060523

**Published:** 2022-06-06

**Authors:** Mirko Magnone, Sonia Spinelli, Giulia Begani, Lucrezia Guida, Laura Sturla, Laura Emionite, Elena Zocchi

**Affiliations:** 1Department of Experimental Medicine, Section of Biochemistry, School of Medical and Pharmaceutical Sciences, University of Genova, Viale Benedetto XV 1, 16132 Genova, Italy; sonia.spinelli@edu.unige.it (S.S.); giulia.begani@edu.unige.it (G.B.); l.guida@unige.it (L.G.); laurasturla@unige.it (L.S.); 2Animal Facility, IRCCS Ospedale Policlinico San Martino, Largo Rosanna Benzi 10, 16132 Genova, Italy; laura.emionite@hsanmartino.it

**Keywords:** ABA, T1D, AMPK, GLUT4, skeletal muscle, LANCL

## Abstract

Abscisic acid (ABA), a plant hormone, has recently been shown to play a role in glycemia regulation in mammals, by stimulating insulin-independent glucose uptake and metabolism in skeletal muscle. The aim of this study was to test whether ABA could improve glycemic control in a murine model of type 1 diabetes (T1D). Mice were rendered diabetic with streptozotocin and the effect of ABA administration, alone or with insulin, was tested on glycemia. Diabetic mice treated with a single oral dose of ABA and low-dose subcutaneous insulin showed a significantly reduced glycemia profile compared with controls treated with insulin alone. In diabetic mice treated for four weeks with ABA, the effect of low-dose insulin on the glycemia profile after glucose load was significantly improved, and transcription both of the insulin receptor, and of glycolytic enzymes in muscle, was increased. Moreover, a significantly increased transcription and protein expression of AMPK, PGC1-α, and GLUT4 was observed in the skeletal muscle from diabetic mice treated with ABA, compared with untreated controls. ABA supplementation in conjunction with insulin holds the promise of reducing the dose of insulin required in T1D, reducing the risk of hypoglycemia, and improving muscle insulin sensitivity and glucose consumption.

## 1. Introduction

Abscisic acid (ABA) is a terpenoid plant hormone that regulates the response of plant tissues to environmental stimuli, both biotic and abiotic, the latter including light exposure, drought, and nutrient availability [1]. ABA is also present and active in animals: in lower Metazoa (marine sponges, hydroids), ABA exhibits a very similar role to a “stress” hormone, regulating sponge respiration and water filtration in response to water temperature, and tissue regeneration in response to light in hydroids [2,3].

As a consequence of its ancient evolutionary origin, before the separation of the plant and animal kingdoms (ABA is also present and active in bacteria, protozoa, and fungi), ABA evolved the capacity to regulate the functional activities of highly specialized mammalian cells in response to diverse environmental “stressors”, the nature of which differs between cell types and tissues. Thus, in mammals, autocrine ABA stimulates the functional response of innate immune cells to mechanical or thermal stimulation (migration, phagocytosis, production of reactive oxygen species and of NO) [4], the activation of microglial cells by bacterial lipopolysaccharide or by β-amyloid [5], of keratinocytes by UV light [6], and the self-renewal of hemopoietic stem cells [7].

More recently, a role for ABA in glycemia control has been unveiled. At nanomolar concentrations, ABA stimulates skeletal muscle (SkM) glucose uptake by increasing the transcription and plasmamembrane translocation of the major glucose transporters expressed in SkM (GLUT4 and GLUT1), and also increases myocyte mitochondrial content and respiration [8]. As a result of the stimulation by ABA of SkM glucose uptake, which occurs both in vivo and ex vivo on the isolated muscle [9], ABA reduces glycemia with an insulin-independent mechanism, reducing insulinemia in the face of a reduced glycemia profile [10]. ABA is present in human plasma, and its concentration increases after a glucose load in normal subjects, but not in patients with T2D or with gestational diabetes [11,12]. Interestingly, resolution of diabetes after childbirth is paralleled by the resumption of a normal plasma ABA increase after glucose load [12].

These results suggest the possible use of ABA to improve glycemic control. The fact that ABA is a plant hormone allows the use of vegetal extracts as a natural source of ABA, also in view of the nanomolar blood concentrations of the hormone required for efficacy. Vegetal extracts titrated in ABA also naturally contain other substances, which may not allow these observed effects to be unequivocally attributed to ABA alone. Indeed, in previous studies, both the pure molecule and an equal dose contained in a vegetal extract were compared and found to be equally effective in reducing glycemia in mammals [10]. Chronic low-dose ABA treatment (1 µg/Kg body weight, BW, for four weeks) improves glucose tolerance, increases muscle glycogen content, and improves physical resistance in mice [9]. The same dose of ABA, chronically administered to subjects with borderline values for prediabetes [13], or who are prediabetic [14], significantly reduces all laboratory parameters employed for diagnosis and follow-up of (pre)diabetes: fasting blood glucose, glycated hemoglobin, HOMA-index, and response to an oral glucose tolerance test (OGTT).

The effect of ABA on skeletal muscle occurs through its receptors LANCL2 and LANCL1 [8]. These proteins belong to the mammalian lanthionine synthase C-like protein (LANCL) family, homologous to the bacterial enzymes that synthesize lanthibiotics, but lacking this catalytic property [15]. The LANCL protein family comprises three members, LANCL3 being expressed at very low levels compared with the other two LANCL proteins [8]. Both LANCL1 and LANCL2 bind ABA, LANCL2 with the highest affinity (Kd values are approximately 3 nM and 1 µM, respectively) [8,16], and overexpression or silencing of these proteins amplifies or reduces the cell responses evoked by ABA [8]. Both LANCL proteins are intracellular: LANCL2 is tethered to the plasma membrane through a myristoyl anchor [17] and, interestingly, shows features typical of receptors of peptide (G-protein linkage) and of steroid hormones (nuclear translocation) [18]. LANCL1 is also a peripheral membrane protein [19], and shares with LANCL2 a significant amino acid sequence identity (>50%) and a similar ubiquitous tissue expression pattern, with the highest expression levels in the brain [8]. Downstream of these receptors, the action of ABA on the SkM involves the activation of the AMPK/PGC-1α/Sirt1 pathway, stimulating the expression and plasma membrane translocation of GLUT1 and of GLUT4, and mitochondrial biogenesis and respiration [8]. In addition to these actions on the SkM, ABA also targets adipose tissue, stimulating glucose uptake and its oxidative consumption, mitochondrial biogenesis and respiratory uncoupling, and expression of browning genes in white adipocytes, in vitro and in vivo, in chronically ABA-treated mice [20].

These observations allow us to hypothesize that ABA evolved as a signal of glucose availability, activating an ancient signaling pathway (AMPK/PGC-1α) and leading to the metabolic utilization of nutrients for energy expenditure. Conversely, insulin (via its master regulatory kinase Akt), is secreted under conditions of high glucose or nutrient availability, and presides over anabolic pathways, leading to energy conservation, principally through protein and triglyceride synthesis. At which point during evolution insulin appeared as a necessary counterbalance to hyperglycemia is unclear; the fact is that, in the absence of insulin, mammalian control of glycemia is severely hampered. However, it should be noted that, both in T1D and in T2D, a severe reduction in endogenous plasma ABA, or of the plasma ABA response to hyperglycemia occur [11]. Thus, the extent to which insulin and ABA synergize to control glycemia is still an open field of investigation. ABA appears to be mainly produced by β-cells, as plasma ABA is very low or undetectable in T1D subjects [11]; thus, a severe reduction in or the complete destruction of β-cells could affect the release of both hormones, which are uniquely endowed with the ability to stimulate muscle glucose uptake.

The aim of this study was to test the hypothesis that, based on the insulin-independent stimulation by ABA of muscle GLUT1 and GLUT4 expression and membrane translocation, treatment with ABA could improve glycemic control in a murine model of T1D. In particular, we aimed to establish: (i) if exogenous ABA could reduce hyperglycemia in multiple low-dose streptozotocin (STZ)-treated mice (i.e., in the presence of markedly reduced endogenous insulin), and (ii) whether ABA could improve the action of suboptimal doses of insulin in single high-dose STZ-treated mice (i.e., in the absence of endogenous insulin). 

In addition, since LANCL1 is overexpressed in the SkM of LANCL2 KO mice, and activates the AMPK/PGC-1α/Sirt1 axis similarly to LANCL2 [8], we compared the effect of ABA on STZ-treated LANCL2−/− (KO) and LANCL2+/+ (wild-type) mice, to verify whether LANCL1 could substitute for LANCL2 in mediating the effect of ABA on the skeletal muscle in diabetic mice.

## 2. Results

### 2.1. Chronic Low-Dose ABA Improves Glycemia in Mice Rendered Diabetic with Multiple-Low Dose STZ

In our experience, a protocol of multiple low-dose STZ injections resulted in a progressively increasing glycemia in treated mice, reaching values of approximately 400 mg/dL after one month. LANCL2+/+ (WT) mice were divided into two groups, one of which was treated with ABA at a dose of 5 µg/Kg BW, administered in drinking water. Starting at day 4, and for 5 consecutive days, both groups of mice were injected i.p. with STZ at a dose of 20 mg/Kg BW/day. A significant reduction in glycemia was observed in the ABA-treated group as compared with ABA-untreated controls, starting from day 14. Glycemia increased during the period when the animals were monitored, reaching 395 ± 64 mg/dL and 294 ± 69 mg/dL at day 28 in the control and ABA-treated groups, respectively (*p* = 0.029) (Figure 1A). At day 30, an OGTT was performed with 0.25 g glucose/Kg BW and glycemia was monitored for 90 min. A reduced elevation of glycemia was observed after gavage in the ABA-treated compared with control mice (Figure 1B), with a significant reduction in the AUC of glycemia over the timespan monitored (Figure 1C). Additionally, when expressed relative to time zero, the glycemia increase after glucose load was significantly reduced in the ABA-treated mice compared with the controls (not shown). In addition to a reduced glycemia profile and an improved response to glucose load, ABA-treated mice lost significantly less body weight compared with ABA-untreated controls (2% as compared with 10%, *p* = 0.02 at day 28, Appendix A), although food intake was not different between the two groups of animals (Appendix A). Peak insulinemia 30 min after glucose gavage was not significantly different in ABA-treated and ABA-untreated mice (0.46 ± 0.25 vs. 0.45 ± 0.14 µg/L), and was approx. 1 log lower compared with peak insulinemia values in STZ-untreated mice undergoing a similar glucose load (3.9 ± 0.7 µg/L; *n* = 3). In line with the plasma insulinemia values, pancreatic insulin mRNA at sacrifice (3 days after the OGTT) was also not significantly different in ABA-treated vs. control mice, and both groups of animals had approx. 1 log lower pancreatic insulin mRNA compared with STZ-untreated mice (not shown).

To explore the effect of chronic ABA treatment started after the onset of hyperglycemia, another experiment was performed: the same protocol of multiple low-dose STZ was applied, but treatment with ABA (5 µg/Kg BW/day) was started at day 8, when mean glycemia was ≥250 mg/dL in both animal groups (Figure 1D).

In the ABA-treated mice, the glycemia profile was again significantly lower compared with the untreated controls, starting from day 13, 5 days after onset of chronic ABA treatment.

At day 30, after treatment with ABA for 22 days, the effect of low-dose insulin on glycemia was explored on ABA-treated and ABA-untreated mice by means of the s.c. injection of 0.1 U of insulin (4 mU/g BW) (Figure 1E). A significant reduction in the glycemia profile after insulin administration was observed in the ABA-treated mice as compared with ABA-untreated mice (Figure 1E), with a significant 50% reduction in the AUC of glycemia in the timeframe 120–240 min (Figure 1F).

Four days after the insulin test, the animals were sacrificed and RT-PCR analysis revealed a significant increase in the transcription of AMPK (6-fold), PGC-1α (2-fold), and GLUT4 (40-fold) in the SkM from ABA-treated mice as compared with ABA-untreated mice (Figure 1G).

Altogether, results shown in Figure 1 indicate that oral ABA, administered either before or after onset of hyperglycemia in a multiple low-dose STZ protocol of diabetes induction, improved the glycemia profile, the response to a glucose load, and the effect of low-dose insulin. When administered after induction of diabetes with low-dose STZ, ABA ameliorated the efficacy of residual endogenous insulin on glycemia, resulting in a significantly reduced glycemia profile in the treated animals (Figure 1D), and also improved the response to exogenous low-dose insulin (Figure 1E). The increased transcription of the AMPK/PGC-1α signaling axis and of GLUT4 in the SkM from ABA-treated mice, compared with untreated controls, confirms previous observations [8] and could be part of the mechanism through which ABA improves glycemia. The increased expression of GLUT4 could also improve the efficacy of residual endogenous insulin on muscle glucose uptake.

### 2.2. A Single Oral Dose of ABA Improves the Efficacy of Insulin in Overtly Diabetic Mice 

To test whether a single oral dose of ABA could improve the effect of insulin on hyperglycemic mice, a protocol of single-high-dose STZ was applied and the effect of ABA administered together with a single dose of insulin was explored at two time points during the development of hyperglycemia. Previous results obtained in humans demonstrated that intake of a single dose of ABA, taken together with a meal, reduced the glycemia profile for up to 6 h, indicating a long-lasting effect of oral ABA [10]. 

After administration of a single high dose of STZ (200 mg/Kg BW), when glycemia was approx. 300 mg/dL in all animals, a first insulin test was performed (Figure 2, upper panel): a group of mice received 0.1 U insulin (4 mU/g BW) s.c. (control) and the other group received the same dose of insulin, together with 5 µg/Kg BW of oral ABA. As shown in Figure 2A, the glycemia profile monitored over 4 h was significantly reduced in the ABA-treated animals as compared with controls, with a significant reduction in the AUC of glycemia, particularly in the timeframe 60–240 min (Figure 2B). A second insulin test (Figure 2, lower panel), similar to the first one, was performed when glycemia was ≥500 mg/dL in all animals (Figure 2C). The same dose of insulin induced a lower decrease in glycemia as compared to the first test, due to the increased basal glucose level. Nonetheless, the glycemia profile observed in the ABA-treated mice was again significantly lower compared with that of control animals (Figure 2C), with a significant reduction in the AUC of glycemia over the 4 h monitoring period after insulin injection (Figure 2D). Again, the most marked reduction in the glycemia profile in the ABA-treated animals was observed in the timeframe 60–240 min. The percentages of reduction in the AUC in the ABA-treated vs. untreated mice were higher in all time frames in the first experiment compared with the second one, and the percentage of AUC reduction observed in the timeframe 120–240 min, i.e., after the maximal effect of insulin was reached (between 60 and 90 min), was 50% in the first experiment and 25% in the second. In both experiments, a significant reduction in the SD of the glycemia values relative to time zero was observed in the ABA-treated group, compared with controls, at all time-points starting from 60 min after insulin injection (*p* = 0.004 and *p* = 0.02 by paired *t* test, respectively).

These results indicate that a single oral dose of ABA, administered together with s.c. insulin, improves glycemic control by prolonging the reduction in glycemia after insulin administration; this effect occurs both under conditions of increased (300 mg/dL) and excess (>500 mg/dL) fasting glycemia, i.e., in the presence or absence, respectively, of residual endogenous insulin production after T1D induction. In addition, the reduced excursion of glycemia values among the ABA-treated mice, compared with controls, indicates a better glycemic control in the first group.

### 2.3. Chronic ABA Treatment Improves the Effect of Insulin in Hyperglycemic T1D Mice

In order to test the effect of chronic ABA treatment on a condition of severe insulin-deficient diabetes, mice were treated with a single high dose of STZ (200 mg/Kg BW, at time zero) and treatment with oral ABA (5 µg/Kg BW/day, at day 5) was started when glycemia was ≥350 mg/dL in all animals. A significant reduction in glycemia was initially observed in the ABA-treated mice (days 5–10); subsequently, when stable hyperglycemia was also reached in the ABA-treated group (≥500 mg/dL), no significant difference between the glycemia profiles of the ABA-treated and ABA-untreated animals was apparent (Figure 3A). This observation suggests that, in the absence of residual insulin secretion, ABA cannot substitute for the hormone, while in the presence of residual insulin secretion, a synergism between the two hormones allows a reduction in glycemia, as already observed after the induction of diabetes with low-dose STZ (Figure 1D).

A first insulin test was performed at day 10, when ABA-treated mice showed a significantly lower glycemia as compared with controls (Figure 3A); two further insulin tests were performed when glycemia was ≥500 mg/dL in all animals, with no significant difference between ABA-treated and ABA-untreated animals. In the three insulin tests, different insulin doses were used (Figure 3A), to test the effect on glycemia of the concomitant administration of a single oral dose of ABA.

In the first test, 0.3 U of insulin (12 mU/g BW) was injected s.c. in all animals; the ABA-treated group also received their daily dose of ABA (5 µg/Kg BW) orally, while the control group received an equal amount of water, and the glycemia profile was monitored until glycemia returned to pre-insulin levels in the control group. In the second and third tests, the dose of insulin was reduced to 0.2 (8 mU/g BW) and 0.1 U (4 mU/g BW), respectively.

With the highest dose of insulin (0.3 U), a significant difference between the glycemia profiles of the ABA-treated and control animals was observed between the time points 90 and 180 min (Figure 3B, upper left panel). The incremental AUC of glycemia was reduced in the ABA-treated animals compared with controls, particularly in the time frame 60–180 min after insulin injection (Figure 3B, upper right panel), and the total AUC over the time span 0–240 min was approx. 30% lower in the ABA-treated mice, as compared with the control mice. When the dose of insulin administered was 0.2 U, the glycemia profile of the ABA-treated vs. ABA-untreated animals was again lower over the time frame 60–180 min (Figure 3B, central left panel), and the total AUC over the time span 0–180 min was reduced by approx. 20% in the ABA-treated mice, as compared with the control mice (Figure 3B, central right panel). Finally, when the dose of insulin was further reduced to 0.1 U, the glycemia profile of the ABA-treated mice was significantly lower than that of controls at each time point after 60 min from insulin administration (Figure 3B, lower left panel). The incremental AUC of glycemia was most significantly reduced in the time frame 60–180 min, and the total AUC over the time span 0–180 min was approx. 40% less than that of controls (Figure 3B, lower right panel).

Collectively, these results indicate that treatment with ABA ameliorated the effect of exogenous insulin in a condition of complete endogenous insulin deficiency, prolonging the timeframe of reduced glycemia over the time span 60–180 min after insulin (and ABA) administration. The highest percentage of reduction in the glycemia AUC between 0 and 180 min (40%) in ABA-treated vs. control mice was observed with the lowest dose of insulin (0.1 U).

In the first insulin test, basal glycemia was significantly different between ABA-treated and ABA-untreated mice, while in the second and third insulin tests basal glycemia was similar in the two groups of mice, indicating that, although treatment with ABA alone did not induce a reduction in glycemia, in combination with exogenous insulin, ABA increased the effect of the hormone (Figure 3B).

Five days after the last insulin test, the mice were sacrificed and basal insulinemia and pancreatic insulin mRNA were measured in all animals. Insulinemia values were not significantly different between ABA-treated and ABA-untreated mice, mean values were 0.22 ± 0.08 and 0.20 ± 0.09 µg/L, respectively, and were approx. 20 times lower than peak insulin levels in STZ-untreated WT mice undergoing a 1 g/Kg BW glucose load to induce transient hyperglycemia.

Pancreatic insulin mRNA also was not significantly different between ABA-treated and ABA-untreated mice and was approx. 200 times lower than in STZ-untreated controls undergoing a 1 g/Kg BW glucose load (Figure 3C).

Although basal glycemia was not significantly different between ABA-treated and ABA-untreated mice at the end point of the experiment (Figure 3A), a statistically significant inverse correlation was observed between basal glycemia and plasma insulin in the ABA-treated mice (Pearson R = −0.86, *p* = 0.001, *n* = 10), but not in the ABA-untreated mice (Pearson R = −0.62, *p* = 0.056, *n* = 10) (Figure 3D). Thus, apparently, on an individual basis, chronic treatment with ABA sensitized mice to the effect of residual endogenous insulin on basal glycemia, although the mean value of glycemia was not significantly different between the two groups of mice. Indeed, the insulin receptor mRNA was approx. 10 times more abundant in the skeletal muscle from ABA-treated mice as compared with control, ABA-untreated mice, and this occurred both in WT and in KO animals (Figure 4F). The increased expression of the insulin receptor in the SkM from ABA-treated mice likely also improved sensitivity to exogenous insulin, contributing to the reduction in glycemia in the ABA-treated mice during the insulin test.

### 2.4. LANCL2 KO Mice Respond to Chronic ABA with a Reduced Glycemia Profile

Although endowed with a somewhat lower affinity for ABA as compared with LANCL2, its homolog LANCL1 also binds ABA and activates the same signaling pathway as LANCL2 in SkM, resulting in the overexpression and plasma membrane translocation of both GLUT1 and GLUT4, via activation of the AMPK/PGC-1α/Sirt1 pathway [8]. LANCL2 KO mice spontaneously overexpress LANCL1 in the SkM (approx. twice the amount of their WT siblings) [8], allowing us to test whether LANCL1 could mediate the beneficial effect of chronic ABA treatment on hyperglycemia induced by low-dose STZ. The same experimental protocol shown in Figure 1 was thus applied in parallel to LANCL2 KO mice: chronic treatment with oral ABA (5 µg/Kg BW/day) was started on day 0, 5 daily consecutive injections of low-dose STZ were performed, starting at day 5 (20 mg/Kg BW/day), and glycemia was monitored for 28 days. At the end point, the mean glycemia of KO mice (294 ± 51 mg/dL) was significantly lower than that of the WT animals (395 ± 64 mg/dL, *p* = 0.029). The glycemia profile of the ABA-treated KO animals was significantly reduced, as compared with that of untreated KO mice, starting from day 14 and until the end of the monitoring period (Figure 4A). In addition, ABA-treated KO mice showed a significantly reduced glycemic response to the final OGTT, performed at day 28, with a smaller increase in glycemia after glucose load (Figure 4B) and, consequently, a decreased AUC of glycemia over the 90 min period of monitoring (Figure 4C). Plasma insulin at 30 min post-gavage was not significantly different between ABA-treated and ABA-untreated LANCL2 KO mice (0.37 ± 0.12 vs. 0.40 ± 0.16 mg/L, respectively) and was similar to that measured in the LANCL2+/+ mice at the end of a similar experimental protocol (Figure 1A). 

The expression levels of glycolytic enzymes were explored in the SkM from both WT and LANCL2 KO animals by RT-PCR on samples from quadriceps muscles taken after sacrifice, 2 days after the OGTT at the end-point of experimentation (Figure 1A and Figure 4A). Transcription of the glycolytic enzymes GaPDH and PFK1, and of the subunit 1 (E1) of the pyruvate dehydrogenase (PDH) complex, was significantly higher in the ABA-treated animals, as compared with the respective untreated controls, with no significant differences between WT and LANCL2 KO mice (Figure 4D), suggesting that ABA stimulates muscle glucose metabolism as well as glucose uptake. mRNA levels of AMPK and PGC-1α were similar in KO (Figure 4E, top) and in WT mice (Figure 1G), while GLUT4 transcription was approx. 2-fold higher in KO as compared with WT mice. Transcription of these genes was however significantly increased in KO mice treated with ABA (Figure 4E, top). As detected by Western blot analysis, SkM from KO mice showed higher protein levels of PGC-1α and GLUT4 than WT controls, which did not further increase upon ABA treatment (excluding AMPK) and were similar to those of WT mice treated with ABA (Figure 4E, center). Finally, an approx. 10-fold stimulation by ABA of the transcription of the insulin receptor mRNA was also observed in LANCL2 KO mice treated with ABA (Figure 4F).

These results indicate that LANCL2 is dispensable for the beneficial effect of exogenous ABA on glycemia control in a diabetic condition induced by low-dose STZ; the effect of ABA in LANCL2 KO mice is likely mediated by (overexpressed) LANCL1, and the dose of exogenously administered ABA is sufficient to activate the signaling pathway downstream of LANCL1, confirming previous results [8].

## 3. Discussion

Therapy for diabetes mellitus is still far from optimal, due to at least two major hurdles: drug administration is discontinuous, and dosage is deliberately suboptimal, to avoid the risk of hypoglycemia. While the first problem is being addressed by technological devices (e.g., continuous glycemia-monitoring systems and insulin-infusion pumps, to improve the timeliness of drug administration), the second is very difficult to overcome. Recent glycemia-controlled infusion pumps do incorporate alarms to alert the patient of incumbent hypoglycemia; however, the risk to the patient is too high for any current therapeutic protocol to attempt to achieve a perfect control of glycemia. Target levels of glycated hemoglobin in diabetic patients are around 7.0% (53 mmol/mol), instead of the normal 5.5% (37 mmol/mol).

Notwithstanding this cautionary attitude by physicians, hypoglycemic events are not rare in diabetic patients, particularly those with T1D, due to their dependence on insulin, arguably the most rapid and potent hypoglycemic drug. T1D patients experience a mean of two episodes of symptomatic hypoglycemia every week and at least one episode of severe hypoglycemia in a year [21]. The inherent risk of hypoglycemia posed by hypoglycemic (including oral) drugs, combined with the high prevalence of diabetes in the population, account for the approx. 100,000 ICU admissions per year in the US alone [22]. Interventions aimed at saving patients with severe hypoglycemia require the concomitant prevention of seizures and of cardiac arrhythmias [21]. Clearly, new strategies to overcome these shortcomings of current diabetes therapy are urgently needed.

Here, we show that administration of ABA together with insulin improves the hypoglycemic effect of low-dose insulin and prolongs its action over time, as compared with an equal dose of insulin alone. In addition, chronic ABA treatment, starting after diabetes induction, improves the glycemia profile in the presence of residual endogenous insulin, and improves the hypoglycemic action of exogenous insulin.

ABA alone cannot substitute for insulin under conditions of total insulin deficiency, when glycemia increases over 500 mg/dL (Figure 3A). However, ABA improves the effect of endogenous insulin when present, but is insufficient to achieve glycemic control (Figure 1A–D and Figure 4A), and ameliorates the effect of exogenous insulin, when the dose of the peptide hormone is insufficient to restore euglycemia (Figure 2A–C and Figure 3B, central left and lower left graphs). These conditions mimic the relative insulin deficiency observed in T2D and the absolute insulin deficiency of T1D, respectively.

Chronic ABA treatment, started before diabetes induction, improves the glycemic profile in treated mice compared with untreated controls during a 28-day period (Figure 1A), without a significant difference between the groups’ plasma insulin levels after a final OGTT, and with similar residual amounts of pancreatic insulin mRNA at the end-point. Thus, the improvement of glycemic control in the ABA-treated animals cannot be attributed to higher endogenous insulin levels, but rather to the glycemia-lowering action of ABA, by means of an increased SkM glucose uptake (Figure 1G and [8]). It is also possible that an increased skeletal muscle expression of the insulin receptor, as observed in the ABA-treated mice in the high-dose STZ protocol (Figure 4F), also occurred in the low-dose STZ experiment, enabling a better response to residual endogenous insulin. 

The signaling pathway activated by ABA in the SkM has been shown to involve the AMPK/PGC-1α/Sirt1 axis, resulting in the increased gene transcription and protein overexpression of the glucose transporters GLUT1 and GLUT4, of the NAD synthesizing enzyme Nampt, of the RabGAP TBC1D1, and of the muscle-specific mitochondrial uncoupling proteins UCP3 and sarcolipin, and in an increased mitochondrial DNA content [8]. Collectively, these transcriptional and translational effects of ABA increase muscle glucose uptake and energy metabolism, leading to increased muscle glucose consumption, as detected in the live rat by microPET [9]. LANCL2, first discovered as a mammalian ABA receptor, and its homolog LANCL1, most recently identified as a second ABA receptor, both activate this signaling pathway [8]. This redundancy of receptors (there is a third member of the LANCL family, LANCL3, whose function remains to be elucidated) may be interpreted as the result of an evolutionary pressure to conserve the action of ABA, underscoring the physiological relevance of this hormone in controlling energy expenditure, not only in the SkM but also in brown and beige adipocytes: ABA stimulates BAT glucose uptake in vivo [20], a measure of BAT thermogenic function [23], and increases transcription of browning genes and mitochondrial biogenesis in beige adipocytes [20]. Interestingly, expression levels of LANCL1 spontaneously increase in muscle cells when the LANCL2 gene is either silenced via interfering RNAs in myoblasts [8], or is absent, as in KO mice [8]. The increased muscle expression of LANCL1 in LANCL2 KO mice, as compared with WT siblings, may explain why LANCL2 KO mice with STZ-induced diabetes respond to ABA similarly to, or perhaps even better than, WT mice (Figure 4A,B). In addition to the increased expression of LANCL1, LANCL2 KO mice have a higher SkM mitochondrial DNA content and increased expression levels of AMPK, PGC-1α, GLUT4/1, Nampt, and UCP3, compared with WT mice, levels which further increase after chronic ABA administration [8]. These transcriptional effects can be attributed to the overexpression of LANCL1, which can substitute for LANCL2 in binding ABA and activating its signaling pathway. Here, we observed a significant increase in the transcription of key glycolytic enzymes (GaPDH, PFK1) and of PDH in the SkM of ABA-treated WT and LANCL2 KO mice, as compared with untreated controls (Figure 4D), which is expected to stimulate oxidative muscle glucose consumption. Indeed, expression levels of LANCL1 were shown to correlate, with an exponential relationship, with mitochondrial O_2_ consumption in LANCL1-overexpressing rat myoblasts [8]. Interestingly, female LANCL2 KO mice (which overexpress LANCL1 in SkM and BAT similarly to male mice) fed a high-glucose diet for 3 months had a significantly lower body weight gain as compared with WT siblings: the body weight relative to time zero was 1.30 ± 0.1 vs. 1.53 ± 0.1 in ABA-treated vs. untreated animals (*n* = 5; *p* < 0.01 by two-tailed, unpaired *t* test) [24].

At the end point of a similar protocol of ABA-pre-treatment followed by diabetes induction with low-dose STZ, the mean glycemia of KO mice (294 ± 51 mg/dL) was significantly lower than that of WT animals (395 ± 64 mg/dL, *p* = 0.001). A possible explanation is the twofold increase in LANCL1 expression in the skeletal muscle of KO compared with WT mice, which may have improved muscle response to exogenous ABA, since LANCL1 shares with LANCL2 the same signaling pathway and stimulatory activity on muscle glucose uptake and energy metabolism [8]. 

Recently, a reduced effect of exogenous ABA on glycemia control was observed in DIO mice bearing a conditional, skeletal muscle ablation of LANCL2 [25]. This mouse model differs from the one used in this study in one important aspect: the DIO mice are a genetic variant of the C57Bl/6 strain, bearing a mutation of the Nicotinamide Nucleotide Transhydrogenase (Nnt) gene, resulting in a truncated and inactive form of this protein, which pumps protons across the inner mitochondrial membrane, providing a means for mitochondrial uncoupling. As a result of this mutation, these mice are highly susceptible to high-fat diets and the development of obesity and insulin resistance [26]. The action of ABA on muscle and on adipose tissue involves the increased transcription and expression of several uncoupling proteins (UCP1, UCP3 and sarcolipin), which is mediated by both LANCL2 and LANCL1 [8,20], making mitochondria one of the main targets of the action of ABA on energy metabolism, increasing glucose consumption by muscle and adipose cells and allowing their sustained glucose uptake. Thus, it is possible that the DIO mutation hampered the uncoupling action of ABA downstream of both LANCL2 and LANCL1, even if LANCL1 was overexpressed in the skeletal muscle of the conditional LANCL2 KO, DIO mice, which was not explored. 

ABA and insulin share several common features: (i) they are produced by pancreatic β-cells, although another source of ABA appears to be the BAT [19], which is likely to be of lesser quantitative importance in humans as compared with rodents; (ii) both stimulate muscle glucose uptake via GLUT4 and glucose metabolism; (iii) they have an ancient evolutionary origin, which can be traced back to unicellular organisms [27,28,29]. However, their metabolic roles are different, as should be expected from their different signaling pathways: ABA stimulates cell (particularly muscle and adipocyte) glucose uptake under conditions of eu- or hyper-glycemia with concomitant energy production (O_2_ consumption and CO_2_ production) [8,20], and, significantly, without stimulating triglyceride synthesis in adipocytes [20]. Meanwhile, insulin stimulates muscle and adipocyte glucose uptake under conditions of hyperglycemia (the stimulus for insulin release from β-cells), and promotes glycogen and triglyceride storage. The metabolic pathways stimulated by insulin are not inhibited by excess ATP; indeed, they are ATP-dependent. Instead, AMPK, the master regulatory kinase in the ABA-signaling pathway, is inhibited by a high ATP/AMP ratio. Additionally, the action of insulin is very fast, but transient, typical of a peptide hormone, while the action of ABA is long-lasting (over several hours after oral intake [10]), in line with its lipophilic chemical structure, endowing it with features typical of a steroid hormone (long half-life, due to its binding to plasma proteins, and intracellular receptors capable of nuclear translocation) [18]. One could speculate that insulin and ABA evolved with a different specific role in the control of energy metabolism: insulin is the hormone in charge of activating the fast biosynthetic responses to transiently abundant nutrient (glucose) availability, and ABA is in charge of stimulating energy and thermogenic metabolism (both in the SkM and BAT) under conditions of sustained, low-level glucose availability.

Chronic ABA treatment has been shown to reduce basal glycemia and improve glucose tolerance after carbohydrate load in subjects with glycemia values borderline with, or within, the limits of prediabetes [13,14,24]. Results presented here suggest that ABA administration could improve glycemic control in the presence of suboptimal insulin therapy, under conditions of reduced or absent endogenous insulin. Several considerations justify future clinical studies to investigate whether ABA supplementation could improve insulin action in T1D patients: together with insulin, plasma ABA is also undetectable or very low in T1D patients [11], suggesting that β-cells are the principal source of endogenous ABA in humans (in rodents, BAT is also a major source of ABA [24]). Thus, the demise of β-cells in T1D greatly reduces the availability of both of the hormones that regulate glycemia, only one of which is currently replaced by therapy. The metabolic actions of insulin and ABA are distinct, thus supplementation with one hormone cannot restore the function(s) of the other. This consideration particularly applies to the “browning” effect of ABA on SkM and on adipocytes. Stimulation by ABA of the expression of uncoupling proteins in the SkM (sarcolipin and UCP3) and in the BAT (UCP1) may be important to increase body energy expenditure, thus contributing to whole-body glucose consumption and the maintenance of euglycemia. Expression of both sarcolipin and UCP3 strongly depend on the expression of the LANCL1/2 proteins, as silencing of both LANCL proteins almost abrogates expression of these muscle-specific uncoupling proteins [8]. ABA does not induce hypoglycemia, even at a dose 100,000 times higher than the one used in this study and in previous clinical studies (100 mg/Kg BW vs. 1 µg/Kg BW); thus, it has a very high therapeutic index (“A new treatment for improving the use of dietary sugar for energy purposes”; Patents EP3716958, US2020376048). The absence of hypoglycemic risk due to excess dosage situates ABA at significant variance with respect to insulin and to oral hypoglycemic drugs. ABA can be administered orally, it is readily absorbed as the protonated molecule in the acidic gastric environment is membrane permeant, and its plasma concentration remains high for several hours after intake [10], probably due to its binding to plasma proteins, which reduce renal clearance. Finally, vegetal extracts rich in ABA have been already used in clinical studies and have shown beneficial effects on glycemia and lipidemia in borderline or prediabetic subjects [10,13,14,25], indicating that nutraceutical products containing ABA at the dose used in this study could provide an immediately available source of the hormone for adjuvant therapy in addition to insulin in T1D patients.

In this preclinical study, the timing of the ABA administration, when given in combination with insulin, was always concomitant; however, in a clinical setting, it might be preferable to postpone ABA administration one to two hours after insulin injection, to maximize its effect in prolonging the efficacy of low dose insulin. Finally, the increased transcription of the insulin receptor mRNA in the skeletal muscle from chronically ABA-treated mice (both WT and LANCL2 KO) suggests that long-term treatment with ABA may improve muscle sensitivity to both endogenous and exogenous insulin. However, the beneficial effect on glycemic control of single-dose ABA in combination with insulin (Figure 2) is evidently not due to transcriptional effects on the insulin receptor, but rather to the ability of ABA to increase glucose uptake and metabolism, synergizing with the action of low-dose insulin.

A still open question requiring further preclinical investigation concerns the possible role on glycemic homeostasis and body weight control of the browning action of ABA on adipose tissue and SkM. Expression of the muscle-specific uncoupling proteins sarcolipin and UCP3 strongly depends on the expression of the LANCL1/2 proteins, as silencing of both LANCL proteins almost abrogates expression of these proteins in rat myoblasts [8]. Moreover, expression levels of LANCL1 correlate with mitochondrial O_2_ consumption in LANCL1-overexpressing rat myoblasts [8]. Finally, LANCL1-overexpressing, LANCL2 KO female mice, fed a high-glucose diet, show a significantly reduced body weight gain as compared with WT siblings, in the face of a higher food intake [24]. These findings suggest that muscle and adipocyte mitochondrial uncoupling and increased oxygen consumption are controlled by the ABA/LANCL1-2 system, and may affect whole-body energy consumption. The extent to which this effect could be advantageous in the diabetic condition remains to be explored. Interestingly, the LANCL1 gene lies within the insulin-dependent diabetes (Idd) 5.3 locus, which provides resistance to T1D in NOD mice [30].

## 4. Materials and Methods

### 4.1. Animals

The LANCL2−/− mouse was generated on a C57Bl/6 background [8]. LANCL2−/− female mice were backcrossed with male C57Bl/6 mice (obtained from Charles River, Milano, Italy) to obtain heterozygous animals. All mice used in this study were derived from a heterozygous LANCL2+/− x LANCL2+/− breeding scheme. Genotyping of the offspring allowed the selection of LANCL2+/+ (henceforth referred to as wild-type, WT) and LANCL2−/− (KO) mice. Mice were housed at the animal facility of the IRCCS San Martino (Genova). All protocols of animal use were approved by the Italian Ministry of Health. For all tests, eight-week old male mice were fed a standard chow diet, and (±)-2-cis, 4-trans abscisic acid (ABA) (Sigma Aldrich, Milano, Italy) was either administered as a single dose by gavage or was administered chronically with the drinking water. In this case, to achieve the required daily dose of ABA (5 µg/Kg BW), the volume of water drunk by the animals was measured 3 times weekly. Based on this volume, and taking into account the average weight of the mice in each cage, the concentration of ABA in the drinking water was adjusted to achieve the required dose.

### 4.2. Diabetes Induction Protocols

STZ was purchased from Cayman Chemical (Michigan, USA) and stored under nitrogen. Solutions were prepared immediately before use in sterile saline, and injected i.p. Two different protocols of diabetes induction were used; multiple low-dose treatment (20 mg STZ/Kg BW/day for 5 consecutive days) or single high-dose treatment (200 mg STZ/Kg BW). Male mice were used throughout the study because female mice are less susceptible to diabetes induction with STZ [31,32]. The multiple low-dose STZ schedule resulted in a progressively increasing glycemia profile over several weeks, allowing us to test the effect of chronic ABA supplementation in the presence of residual endogenous insulin. In contrast, the single high-dose schedule of STZ administration caused a very rapid development of hyperglycemia (>500 mg/dL) over a few days, and allowed us to test the effect of ABA in a condition of complete insulin deficiency.

Glycemia was measured with a glucometer (Ascensia, Milan, Italy) on blood droplets taken from the tail vein, and BW was monitored 3 times weekly.

### 4.3. Oral Glucose Tolerance Test (OGTT)

Mice were fasted for 6 h before the OGTT. In view of the condition of increased basal glycemia caused by STZ treatment, the dose of glucose was reduced from the usual 1 g/Kg BW to 0.25 g/Kg BW, administered by gavage in a 150-µL water solution, with or without ABA (5 µg/Kg BW). Blood was drawn from the tail vein before gavage (time zero) and 15, 30, 60, and 120 min after gavage. Glycemia was measured with a glucometer (Ascensia, Milan, Italy) on blood droplets taken from the tail vein and each measurement was performed in duplicate. The area-under-the-curve (AUC) of glycemia was calculated with the trapezoidal rule from the values of glycemia (in mg/dL) (absolute AUC), or from the glycemia values relative to time zero (incremental AUC).

### 4.4. Insulin Tests

Insulin (Toujeo, Sanofi-Aventis, Paris, Fance) was diluted immediately prior to use in a mildly acidic aqueous solution, and injected subcutaneously. When given together with insulin, ABA (5 µg/Kg BW) was administered by gavage immediately before insulin injection. Control, ABA-untreated mice received an equal amount of water by gavage.

For determination of plasma insulin levels, blood samples taken from the tail vein or from the orbital sinus of mice were centrifuged, and plasma was immediately frozen in dry ice. The insulin concentration was measured with a high-sensitivity ELISA kit (ultrasensitive insulin ELISA, Mercodia, Winston Salem, NC, USA).

### 4.5. qPCR Analysis

After sacrifice, mouse quadriceps samples (approx. 30 mg) were immediately frozen in liquid nitrogen for subsequent analysis. Total RNA was extracted from the muscle using QIAzol Lysis Reagent and Tissue Lyser instrument (Qiagen Italia, Milano, Italy), according to the manufacturer’s instructions. The cDNA was synthesized by using iScript cDNA Synthesis Kit (Bio-Rad Laboratories, Milano, Italy), starting from 1 μg of total RNA, and was used as a template for qPCR analysis: reactions were performed in an iQ5 Real-Time PCR detection system (Bio-Rad Laboratories, Milano, Italy) as described [33]. The mouse-specific primers were designed using Beacon Designer 2.0 software (Bio-Rad Laboratories, Milano, Italy), and their sequences are listed in Appendix A). Statistical analysis of the qPCR was performed using the iQ5 Optical System Software version 1.0 (Bio-Rad Laboratories, Milano, Italy) based on the 2^−ΔΔCt^ method [34]. Values for mouse genes were normalized on hypoxanthine-guanine phosphoribosyltransferase-1 mRNA expression. To verify the purity of the products, a melting curve was produced after each run. The dissociation curve for each amplification was analyzed to confirm the absence of nonspecific PCR products.

### 4.6. Western Blot

After the explant, quadriceps samples isolated from wild type (WT) or LANCL2−/− mice (KO), either treated or not treated with 100 nM ABA, were lysed with Tissue Lyser (Qiagen Italia, Milano, Italy) and centrifuged for 10 min at 12,000× *g*, and the supernatants were analyzed by Western blot. Lysates (70 µg) were loaded on 10% polyacrylamide gel and separated by SDS-PAGE, and proteins were transferred to nitrocellulose membranes (Bio-Rad Laboratories, Milano, Italy), according to standard procedures. The membranes were blocked for 1 h with 20 mM Tris-HCl pH 7.4, 150 mM NaCl, 1% Tween 20 (TBST) containing 5% non-fat dry milk and incubated for 1 h at room temperature with primary antibodies (anti-PGC-1α, anti-AMPK, anti-GLUT4 and anti-Vinculin, Appendix A). Following incubation with the appropriate secondary antibodies and ECL detection (GE Healthcare), band intensity was quantified with the ChemiDoc imaging system (Bio-Rad Laboratories, Milano, Italy).

### 4.7. Statistical Analysis

The normal distribution of the values obtained from the murine experiments was assessed with the Vassarstats website for statistical computation (VassarStats: Website for Statistical Computation. http://www.vassarstats.net 13 August 2021). Continuous variables are presented as mean ± SD. Comparisons were drawn by an unpaired, two-tailed *t*-test, and statistical significance was set at *p* < 0.05. In the legends to the figures, the calculated *p* values are indicated as follows: * *p* < 0.05, # *p* < 0.02, and $ *p* < 0.005. The Pearson correlation was used to analyze two sets of data (glycemia and insulinemia) in Figure 3D.

## 5. Conclusions

In conclusion, these results indicate that oral ABA at 5 µg/Kg, given daily for 30 days, reduces the progressive increase in glycemia in mice treated with low-dose STZ. When administered as a single dose together with insulin, ABA increases the effect of insulin in mice treated with high-dose STZ. Increased transcription of AMPK, PGC-1α, GLUT4, key glycolytic enzymes, and of the insulin receptor occur in the skeletal muscle of ABA-treated diabetic mice. These results warrant clinical studies, aimed at evaluating the potential benefit of oral ABA supplementation to ameliorate glycemic control in insulin-deficient, as well as in insulin-resistant, patients. In insulin-deficient patients, ABA supplementation should improve and prolong the action of exogenous insulin, perhaps allowing them to reduce their daily dose of insulin (and the risk of hypoglycemia), while at the same time ameliorating glycemic control. In insulin-resistant patients, ABA supplementation could contribute to the reduction in post-prandial hyperglycemia, in conjunction with oral hypoglycemic drugs, by synergizing with the action of endogenous insulin, via a different signaling pathway, eliciting an increased muscle glucose uptake.

## Figures and Tables

**Figure 1 metabolites-12-00523-f001:**
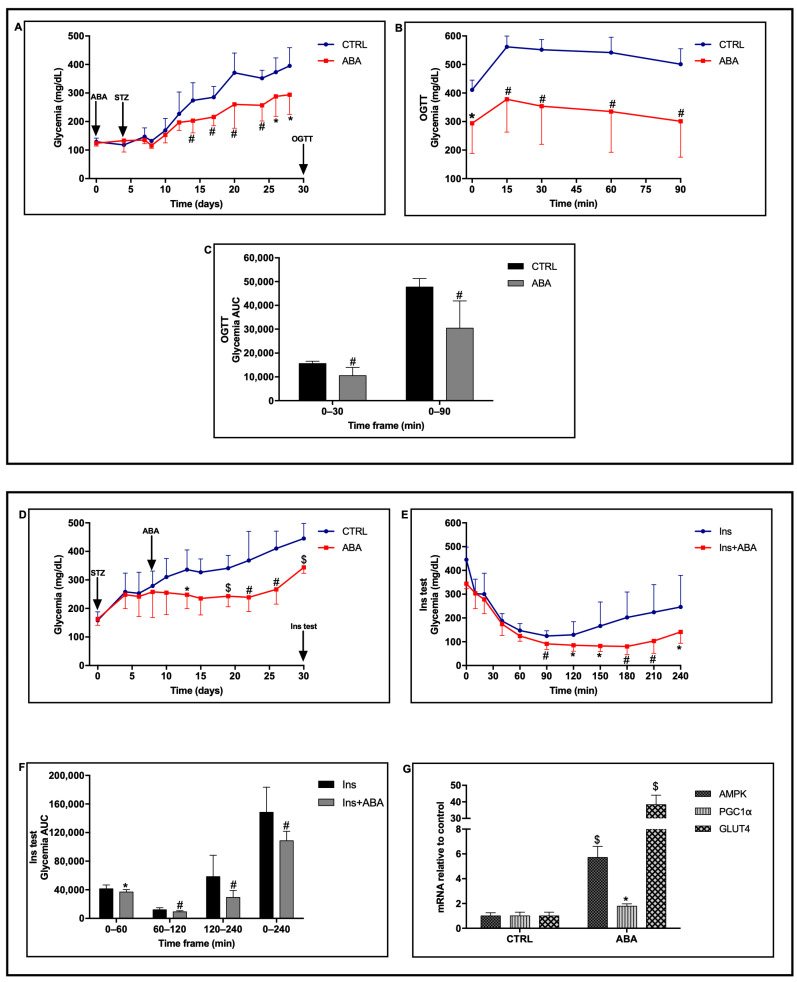
Chronic ABA treatment in a multiple low-dose STZ protocol reduces glycemia and improves glucose tolerance and the effect of exogenous insulin. Eight-week old male LANCL2+/+ (wild-type) mice, obtained from a heterozygous breeding scheme (see Materials and Methods) were divided into two groups (8 mice/group) and treated or not (control, CTRL) with ABA at 5 µg/Kg BW/day starting either before (panels (**A**–**C**)) or after (panels (**D**–**G**)) diabetes induction with multiple low-dose streptozotocin (STZ) at 20 mg/Kg BW/day for 5 days. Upper panel: (**A**) glycemia profile over 30 days of control and of ABA-treated mice, ABA treatment starting at day 0 followed by STZ treatment starting at day 4 (arrows). (**B**) glycemia profile during the OGTT, performed at day 30 on the same animals with 0.25 g glucose/Kg BW, without (control) or with ABA (5 µg/Kg BW). (**C**) AUC of the glycemia profile of the OGTT. Lower panel: (**D**) glycemia profile over 30 days of control and of ABA-treated mice, ABA treatment starting 3 days after completion of STZ treatment (arrows). (**E**) glycemia profile of the insulin test performed at day 30 with 0.1 U of insulin (Ins), injected s.c. at time zero. (**F**) AUC of the glycemia profile of the insulin test. (**G**) mRNA relative to control of AMPK, PGC-1α and GLUT4 in the skeletal muscle from control and ABA-treated mice sacrificed 4 days after the insulin test. * *p* < 0.05, # *p* < 0.02 and $ *p* < 0.005 relative to CTRL or Ins. *p* values are calculated by unpaired two-tailed *t*-test.

**Figure 2 metabolites-12-00523-f002:**
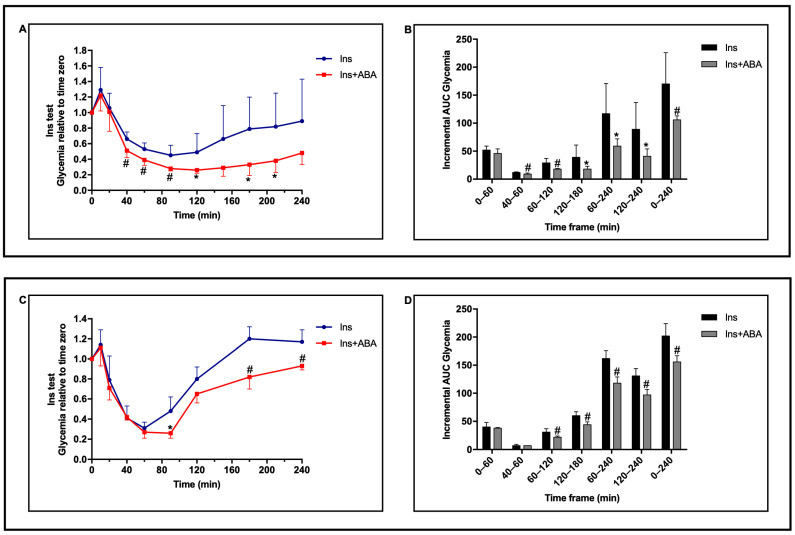
A single dose of oral ABA in addition to insulin improves glycemia control in hyperglycemic mice. LANCL2+/+ (wild-type) mice (8/group) received a single high-dose STZ injection (200 mg/Kg BW). When glycemia was >300 mg/dL in all animals, a first insulin test was performed (upper panel) with 0.1 U of insulin injected s.c. without (Ins) or with (Ins + ABA) the concomitant administration of 5 µg/Kg BW of ABA by gavage (**A**,**B**). A similar insulin test was repeated when glycemia was >500 mg/dL in all animals (lower panel, (**C**,**D**)). (**A**,**C**): glycemia profile during the insulin test. (**B**,**D**): incremental AUC of glycemia in the respective insulin test shown in the opposite panel (panel (**A**) or (**C**)), as calculated in the indicated timeframes. * *p* < 0.05 and # *p* < 0.02 relative to Ins. *p* values are calculated by unpaired two-tailed *t*-test.

**Figure 3 metabolites-12-00523-f003:**
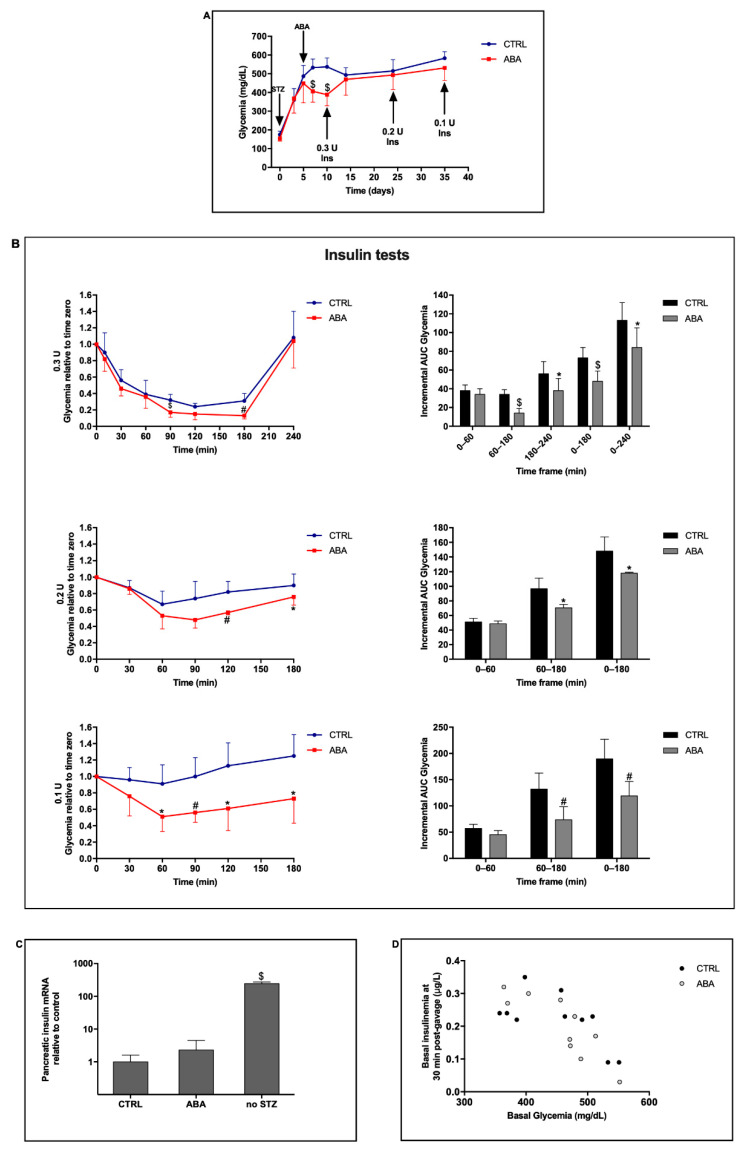
Chronic ABA treatment improves the efficacy of low-dose insulin in hyperglycemic T1D mice. LANCL2+/+ (wild-type) mice (10/group) received a single high-dose STZ injection (200 mg/Kg BW) at time zero, and oral ABA treatment (5 µg/Kg BW/day) was started at day 5. Three insulin tests were performed at days 10, 24, and 35 (arrows) by injecting s.c. 0.3, 0.2 or 0.1 U insulin, respectively, without (control, CTRL) or with the daily oral dose of ABA. Panel (**A**): glycemia profile of the mice and timing of the treatments with STZ and ABA (arrows). Panel (**B**) shows results of the insulin tests: left panels glycemia profile in control (CTRL) and ABA-treated mice (ABA); right panels AUC of glycemia of the corresponding insulin test. Five days after the last insulin test, mice were sacrificed and basal glycemia and insulinemia and pancreatic insulin mRNA were measured. Panel (**C**): pancreatic insulin mRNA of ABA-treated and of STZ-untreated mice relative to control, ABA-untreated mice. Panel (**D**): Pearson correlation between basal glycemia and insulinemia in ABA-untreated (CTRL) and ABA-treated mice. * *p* < 0.05, # *p* < 0.02 and $ *p* < 0.005 relative to CTRL. *p* values are calculated by unpaired two-tailed *t*-test. The Mann–Whitney U test (one-tailed) was used only for the end-point of the glycemia profile in Panel (**B**), central, left panel, as the two sets of data were not normally distributed.

**Figure 4 metabolites-12-00523-f004:**
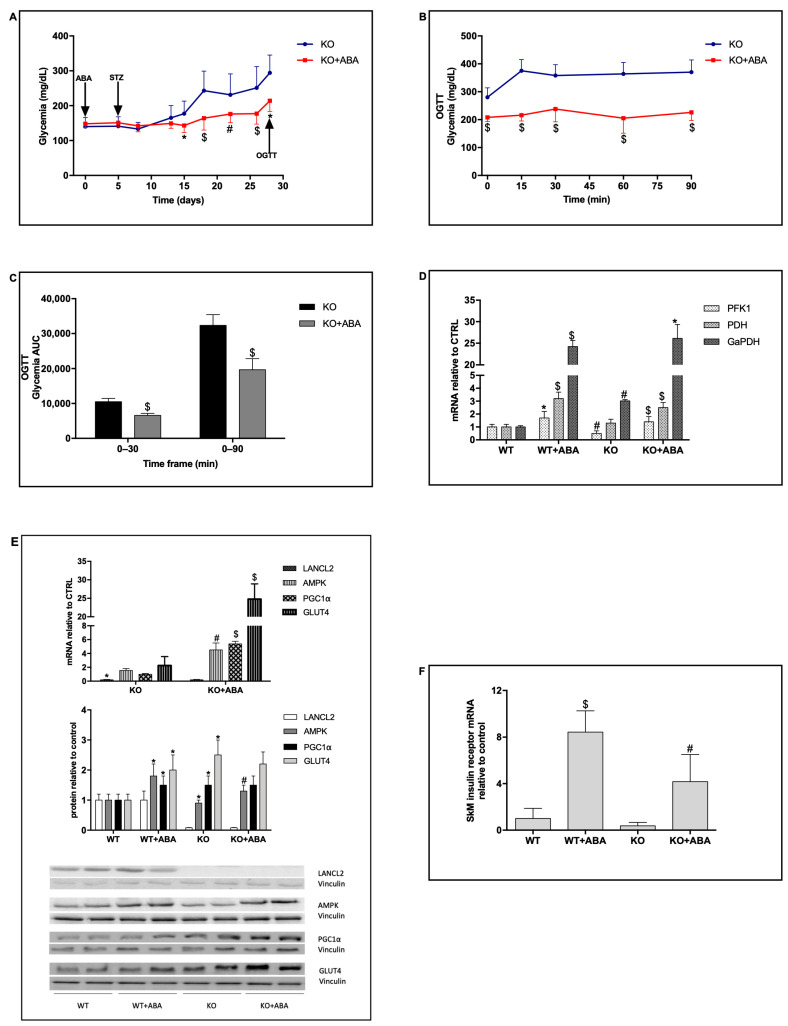
Chronic ABA treatment improves the glycemia profile in LANCL2 KO mice and increases transcription of glycolytic enzymes in the skeletal muscle. Eight-week old LANCL2−/− (KO) male mice obtained from the same heterozygous breeding scheme as in Figure 1 were divided into two groups (9 mice/group) and treated, or not, with oral ABA (5 µg/Kg BW/day) starting at day 0. Starting five days after initiation of ABA treatment, all mice received five daily consecutive s.c. injections of streptozotocin (STZ) at 20 mg/Kg BW. At day 28, all animals received an OGTT with 0.25 g glucose/Kg BW. Panel (**A**): glycemia profile of KO mice, treated with ABA (KO + ABA), or untreated (KO). The arrows indicate the timing of the various treatments. * *p* < 0.05, # *p* < 0.02 and $ *p* < 0.005 relative to KO. Panel (**B**): glycemia profile of the OGTT. $ *p* < 0.005 relative to KO. Panel (**C**): AUC of glycemia in the indicated timeframes after the OGTT. $ *p* < 0.005 relative to KO. Panel (**D**): mRNA levels of glycolitic enzymes in the skeletal muscle of LANCL2+/+ (wild-type, WT) and LANCL2−/− (KO) mice after sacrifice (day 30). GaPDH, glyceraldehyde 3-phosphate dehydrogenase, # *p* < 0.02 and $ *p* < 0.005 relative to WT and * *p* < 0.05 relative to KO; PFK1, phosphofrutto kinase, * *p* < 0.05 and # *p* < 0.02 relative to WT and $ *p* < 0.005 relative to KO; PDH, subunit 1 of pyruvate dehydrogenase, $ *p* < 0.005 relative to WT and KO. Panel (**E**): top, mRNA levels of PGC-1α, AMPK and GLUT4 in the skeletal muscle of LANCL2−/− (KO) mice relative to control, ABA-untreated mice (CTRL) of Figure 1G; values are the mean ± SD from *n* = 9 animals; center, protein expression of LANCL2, PGC-1α, AMPK and GLUT4 is expressed relative to levels in untreated WT muscle; values are the mean ± SD from *n* = 5 animals. Bottom, a representative Western blot analysis is shown for *n* = 2 animals for each condition. Values are normalized against Vinculin, as housekeeping protein. * *p* < 0.05 relative to WT and # *p* < 0.02 and $ *p* < 0.005 relative to KO. Panel (**F**), insulin receptor mRNA in the SkM of LANCL2+/+ (wild-type, WT) and LANCL2−/− (KO) mice, $ *p* < 0.005 relative to WT and # *p* < 0.02 relative to KO. *p* values are calculated by unpaired two-tailed *t*-test.

## Data Availability

Not applicable.

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
