# Peer review of "Abscisic Acid Improves Insulin Action on Glycemia in Insulin-Deficient Mouse Models of Type 1 Diabetes"

_metabolites, 2022, doi:10.3390/metabo12060523_

Round 1

Reviewer 1 Report

Summary

This manuscript reported the therapeutic effects of oral administration of abscisic acid (ABA) on glycemia in type 1 diabetes (T1D) model mice. Not only ABA alone reduces blood glucose, ABA improved insulin action in oral glucose tolerance test (OGTT). The beneficial effects of ABA observed in knockout mice of LANCL2, a receptor of ABA, suggests that another receptor LANCL1 might be involved in ABA action. ABA upregulated the expression of AMPK, PGC1α, and GLUT4 in skeletal muscle, which should be a part of the mechanism of action of ABA.

The animal experiments appropriately performed and the data indicated the therapeutic effects of ABA on T1D glycemia. Some supplemental data and information commented below will make perfect the study.

Comments

1.          Line 232: Body weight and food intake should be shown by table or graph in supplementary data or somewhere. It’s very important information to provide the safety of ABA. Were there no differences the weights of fats and muscles?

2.          LANCL2 expression in KO and control mice should be verified by qPCR or Western and the data must be displayed.

3.          Figure 3D and Line 402: “Significant inverse correlation” is better to be highlighted more in Figure 3D. The range of X-axis will be good to be changed to 300-600.

Minor points

4.          Line 13: “T1D” should be defined.

5.          Line 45: In the muscular field, “SM” usually means “smooth muscle”. In general, abbreviation of skeletal muscle is “SkM”.

6.          Line 79: Does “sequence identity” mean amino acid seq or DNA seq?

7.          Line 200: Information of the antibodies (manufacturer, CatNo., etc.) should be described.

Author Response

Reviewer 1

Comments

  1. Line 232: Body weight and food intake should be shown by table or graph in supplementary data or somewhere. It’s very important information to provide the safety of ABA. Were there no differences the weights of fats and muscles?

A new Figure has been added as Supplementary Materials (Figure 1S), containing data on body weight and food intake. Unfortunately, we do not have data regarding the weights of fat and muscle.

  1. LANCL2 expression in KO and control mice should be verified by qPCR or Western and the data must be displayed.

These data have been added to Figure 4E.

  1. Figure 3D and Line 402: “Significant inverse correlation” is better to be highlighted more in Figure 3D. The range of X-axis will be good to be changed to 300-600.

The X-axis in Figure 3D has been changed as suggested.

Minor points

  1. Line 13: “T1D” should be defined.

T1D has been defined in the abstract.

  1. Line 45: In the muscular field, “SM” usually means “smooth muscle”. In general, abbreviation of skeletal muscle is “SkM”.

SM has been changed into SkM throughout the manuscript.

  1. Line 79: Does “sequence identity” mean amino acid seq or DNA seq?

The sequence identity has been specified as amino acid identity (line 81).

  1. Line 200: Information of the antibodies (manufacturer, CatNo., etc.) should be described.

A new Table has been added to the Supplementary Materials (Table 2), containing this information.

Reviewer 2 Report

Authors stating that, administration of Abscisic acid (ABA), a terpenoid plant-based hormone regulate the glucose homeostasis and improve the insulin sensitivity in muscle via insulin independent manner in a mouse model of type 1 diabetes (T1D). Several previous invivo and exvivo studies have been shown that ABA can stimulates the glucose uptake in skeletal muscle by increasing both transcript and translocation of glucose transporters to plasma membrane suggesting the beneficial effects of ABA under hyperglycemic condition. Based on several published reports, the authors claimed the novelty of the current study is ABA treatment improved the hyperglycemia via insulin-independent manner by stimulating the GLUT 1 and GLUT 4 transporters in skeletal muscle of a murine model of T1D.  In addition, study using LANCL2 KO mice showing the direct stimulation of ABA in skeletal muscle mediated by its receptors. However, authors need address the several comments before the manuscript accepted by Metabolites, a peer reviewed journal of MDPI.

Major Comments:

1.      In Material and Methods, authors never mentioned or not clear the start point of the ABA treatment and what is the terminal time point of the experiment? When the mice were sacrificed?

2.      Also, when did authors performed the OGTT? At the end of the study period or during?

3.      In Figure 1B, authors need to mention on the subject tile line that it is OGTT and have to indicate clearly in the Figure legend.

4.      Figure 1, Figure legends need to rewrite and have to indicate clearly with respective representative figure.

5.      Similar changes need to modify for Figure 2 along with Figure legend.

Minor comments:

1.      Line 170-171, Adjust the format for sub-title of qPCR analysis

2.      Line 187, authors need to show the Table of the Primer sequences as Supplementary method of the current manuscript. No need to add the Table 1 in the Material and methods section.

Author Response

Reviewer 2

Comments and Suggestions for Authors

Authors stating that, administration of Abscisic acid (ABA), a terpenoid plant-based hormone regulate the glucose homeostasis and improve the insulin sensitivity in muscle via insulin independent manner in a mouse model of type 1 diabetes (T1D). Several previous invivo and exvivo studies have been shown that ABA can stimulates the glucose uptake in skeletal muscle by increasing both transcript and translocation of glucose transporters to plasma membrane suggesting the beneficial effects of ABA under hyperglycemic condition. Based on several published reports, the authors claimed the novelty of the current study is ABA treatment improved the hyperglycemia via insulin-independent manner by stimulating the GLUT 1 and GLUT 4 transporters in skeletal muscle of a murine model of T1D.  In addition, study using LANCL2 KO mice showing the direct stimulation of ABA in skeletal muscle mediated by its receptors. However, authors need address the several comments before the manuscript accepted by Metabolites, a peer reviewed journal of MDPI.

Major Comments:

  1. In Material and Methods, authors never mentioned or not clear the start point of the ABA treatment and what is the terminal time point of the experiment?

Indeed, each single experiment has its own different schedule of treatment and timing of sacrifice, making it difficult to generalize a procedure in the Methods section.  For this reason, in the Materials and Methods the general procedures are described, while specific details regarding each single experiment were given in the legends and also in the Results (now highlighted in red in the manuscript). To improve clarity at first glance, the Figures have now been modified: the starting point of ABA treatment is indicated by an arrow (Fig. 1A, D, Fig. 3 A and Fig. 4A), as is the starting point of the STZ treatment in the same panels; the timing of the OGTTs and of the insulin tests were already indicated in the legends (Figs. 1A and Fig. 4A), but have also been added directly in the figures as arrows (Fig. 1A, D, Fig. 3A and Fig. 4A); the y-axis of Figs. 1B, 1E, 1F, 2A, 2C, 3B graphs on the left, 4B, 4C has been modified to indicate the test performed

When the mice were sacrificed?

This info was already indicated in the legends and in the text and has now been highlighted.

  1. Also, when did authors performed the OGTT? At the end of the study period or during?

The timing of the OGTT was indicated in the legends (now highlighted); a new arrow has been added to Figs. 1A, and 4A.

  1. In Figure 1B, authors need to mention on the subject tile line that it is OGTT and have to indicate clearly in the Figure legend.

The Y-axis in Fig. 1B has been modified as suggested; the legend already indicated that panel B referred to the glycemia profile after OGTT (text highlighted).

  1. Figure 1, Figure legends need to rewrite and have to indicate clearly with respective representative figure.

New boxes have been created to include panels regarding the same experiment in Fig. 1. The legend has been modified accordingly and the timings of all treatments have been highlighted.

  1. Similar changes need to modify for Figure 2 along with Figure legend.

Similar changes have been applied to Fig. 2 and to its legend.

Minor comments:

  1. Line 170-171, Adjust the format for sub-title of qPCR analysis

The format has been adjusted (new line 646); Materials and Methods were translocated at the end of the Discussion as requested by the Journal.

  1. Line 187, authors need to show the Table of the Primer sequences as Supplementary method of the current manuscript. No need to add the Table 1 in the Material and methods section.

The Table has been translocated in the Supplementary Materials.
